# When Action Speaks Louder than Words: Exploring Non-Verbal and Paraverbal Features in Dyadic Collaborative VR

**DOI:** 10.3390/s25175498

**Published:** 2025-09-04

**Authors:** Dennis Osei Tutu, Sepideh Habibiabad, Wim Van den Noortgate, Jelle Saldien, Klaas Bombeke

**Affiliations:** 1imec-mict-UGent, Department of Communication Sciences, Ghent University, Miriam Makebaplein 1, 9000 Ghent, Belgium; sepideh.habibiabad@ugent.be (S.H.); klaas.bombeke@ugent.be (K.B.); 2 imec-itec-KULeuven, Department of Psychology and Educational Sciences, KU Leuven, Etienne Sabbelaan 51, 8500 Kortrijk, Belgium; wim.vandennoortgate@kuleuven.be; 3 imec-mict-UAntwerpen, Department of Product Development, University of Antwerp, Paardenmarkt 94, 2000 Antwerp, Belgium; jelle.saldien@uantwerpen.be

**Keywords:** soft skills training, communication, collaboration, multimodal behavior tracking, VR interaction analysis

## Abstract

Soft skills such as communication and collaboration are vital in both professional and educational settings, yet difficult to train and assess objectively. Traditional role-playing scenarios rely heavily on subjective trainer evaluations—either in real time, where subtle behaviors are missed, or through time-intensive post hoc analysis. Virtual reality (VR) offers a scalable alternative by immersing trainees in controlled, interactive scenarios while simultaneously capturing fine-grained behavioral signals. This study investigates how task design in VR shapes non-verbal and paraverbal behaviors during dyadic collaboration. We compared two puzzle tasks: Task 1, which provided shared visual access and dynamic gesturing, and Task 2, which required verbal coordination through separation and turn-taking. From multimodal tracking data, we extracted features including gaze behaviors (eye contact, joint attention), hand gestures, facial expressions, and speech activity, and compared them across tasks. A clustering analysis explored whether o not tasks could be differentiated by their behavioral profiles. Results showed that Task 2, the more constrained condition, led participants to focus more visually on their own workspaces, suggesting that interaction difficulty can reduce partner-directed attention. Gestures were more frequent in shared-visual tasks, while speech became longer and more structured when turn-taking was enforced. Joint attention increased when participants relied on verbal descriptions rather than on a visible shared reference. These findings highlight how VR can elicit distinct soft skill behaviors through scenario design, enabling data-driven analysis of collaboration. This work contributes to scalable assessment frameworks with applications in training, adaptive agents, and human-AI collaboration.

## 1. Introduction

In today’s professional and educational settings, soft skills—such as communication and collaboration—are increasingly recognized as critical for success [1,2,3,4], yet they remain difficult to measure, are complex to teach, and elusive to assess [5]. Traditional training methods—workshops, team exercises, and role-playing scenarios—attempt to cultivate these skills by placing individuals in controlled interpersonal settings. However, these approaches often fragment interaction into isolated moments, stripping communication of its natural flow. Soft skills are more than words exchanged; they are glances, pauses, and gestures—the silent choreography of human interaction [4,6].

Attempts to capture these nuances have often relied on self-reports, peer evaluations, and observational rubrics. However, such assessments are prone to bias, subjectivity, and inconsistent interpretation [7,8,9]. Even structured tools—like communication rubrics—are limited to what observers can consciously detect and report [6]. Many micro-level cues—such as shifts in gaze, silent turn-taking, or the timing of a pause—form the subtle architecture of non-verbal communication, yet often remain overlooked in traditional assessments. To study non-verbal cues, researchers have relied heavily on lab-based technologies like eye-tracking cameras and facial recognition software. These tools have also been employed to quantify behaviors and emotional states [10,11,12,13]. However, these assessments are often limited by their highly controlled environments, where static positioning and intrusive sensors can disrupt natural social dynamics. The Partnership for 21st Century Learning (P21) framework underscores the importance of interpersonal skills in preparing individuals for today’s workforce, highlighting the need for robust and scalable methods of training and assessment that go beyond subjective evaluations [14]. This is where we believe virtual reality (VR) could play a major role.

Virtual reality (VR) is increasingly used in education and training for its ability to simulate realistic, risk-free environments. In medical education, VR reduces injuries, speeds up procedures, and improves outcomes by allowing learners to practice safely and repeatedly. Nursing students report that VR is immersive and helpful for building confidence and practicing without risking patient harm [15]. In engineering, students explore complex systems like thermal labs through VR, gaining faster insights than in traditional settings [16]. High-risk industries, such as construction, use VR for safety training. It helps workers memorize hazards and practice procedures in engaging, repeatable ways [17]. These examples shows VR’s value in offering embodied, scenario-based learning. Still, VR is best seen as a supplement—not a replacement—for real-world practice, especially for tasks requiring fine motor skills [15].

### 1.1. The Potential of VR for Soft Skill Training

Unlike physical training settings, VR provides a controlled yet dynamic medium where both verbal, non-verbal, and paraverbal signals can be systematically captured in real time. Advances in VR tracking systems now allow researchers to record eye gaze patterns, gestures, and expressions, offering an objective, sensor-driven analysis of human interaction [18,19,20]. Some studies have focused on facial emotion recognition in VR environments, highlighting the potential for real-time affective feedback but also revealing gaps in defining precise thresholds for interpreting subtle facial muscle movements [21,22]. Others demonstrated how multimodal features such as gaze, posture, and facial cues can assess learning concentration in VR, linking behavioral signals with cognitive engagement [23]. In fact, VR-based assessments could offer the potential to move beyond purely subjective interpretation by enabling the measurement of observable behaviors [24]. However, translating these observable cues into underlying soft skills is far from straightforward. This process still requires human interpretation. Although sensor data can capture observable behaviors, the mapping of these behaviors to underlying soft skills is not inherently objective. Until reliable models exist that can infer latent constructs from such data, we remain dependent on human-labeled interpretations—interpretations shaped by subjective judgments and context-specific norms. Current commercial VR platforms, such as Bodyswaps, exemplify this dependency. Bodyswaps primarily offers training scenarios that involve user interaction with AI-driven virtual agents, focusing heavily on verbal user inputs [25]. While these platforms provide real-time behavioral feedback based on expert-driven models, their interpretations of user behavior remain fundamentally anchored in human-labeled examples and subjective criteria.

Rather than treating virtual interaction as a mere simulation of reality, we consider it a domain in its own right—an emergent space where behavioral transfer and adaptation unfold [19]. In such environments, it is not only possible but also likely that new forms of interaction—behaviors unique to or amplified by the VR medium—will emerge. Platforms like VRChat already demonstrate how users develop distinct social norms, body language, and interaction patterns tailored to avatar-based immersive contexts [26]. Despite these possibilities, existing research in VR-based communication training remains limited in scope [6,27]. Prior studies have described VR interactions but have not systematically examined how scenario design—through specific design choices and embedded limitations—shapes the elicitation of interaction behaviors. While it is well understood that verbal coordination will depend on the structural features of a task, the influence of scenario design on the emergence of non-verbal and paraverbal dynamics remains underexplored. Therefore, the aim of this study is to explore which sensor features reflecting communicative and collaborative behavior emerge with certain task affordances and whether or not these features can be clustered. Doing so will allow us to build better VR scenarios for soft skill assessment in future work.

In fact, in the last decade, virtual reality (VR) has rapidly evolved into a versatile tool for studying and training complex human skills, including communication, collaboration, and more [27]. One significant application has been immersive role-play simulations designed to replicate social interactions under controlled conditions. For instance, recent research explores how AI-driven conversational agents can create adaptive, realistic dialogues for training scenarios ranging from social emotion elicitation to conflict resolution [28,29]. Platforms like Bodyswaps leverage similar principles, offering users the chance not only to engage in virtual role-play but also to experience interactions from new perspectives, such as reviewing conversations from the viewpoint of an AI agent or a different avatar [30,31,32]. Such perspective-shifting techniques have shown promise in increasing presence and empathy, underscoring VR’s unique affordances compared to traditional training approaches.

### 1.2. Current Study

Two VR tasks were developed to elicit contrasting communication strategies in collaborative scenarios by embedding intentional design mechanisms. Grounded in the Mechanics–Dynamics–Aesthetics (MDA) framework [33] and cooperative learning theories [34,35,36,37,38], the tasks leverage design choices that subtly guide user interaction. Rather than instructing participants explicitly, these scenarios were crafted using mechanics inspired by these frameworks—such as shared or limited visibility, interdependent roles, and enforced sequencing—to influence how collaboration unfolds. The MDA framework provides a lens for connecting low-level mechanics (e.g., restricting object visibility or controlling access to interaction zones) to higher-level user behaviors and experiences. Within educational VR, this helps shape how learners engage with both the environment and each other. Design insights from escape room literature further support this strategy, emphasizing how challenge escalation, asymmetric information, and forced cooperation can increase engagement and prompt collaborative problem-solving [35,36]. These structures naturally encourage communication strategies such as verbal clarification, coordination, and collective decision-making. In addition, research on organizational collaboration shows that structured workflows and role-based dependencies foster sustained interaction—especially when tasks require coordination under limited visibility or constrained access to shared goals [34]. Gamification studies also suggest that goal interdependence and real-time feedback mechanisms can heighten engagement while influencing the tone and tempo of cooperative behavior [37,38].

In this study, these design principles are applied to multi-user VR scenarios with the specific goal of investigating soft skills in action. Instead of focusing solely on participant choices, the environment itself shapes how collaboration emerges. Carefully embedded constraints and affordances create conditions that draw out behaviors such as negotiation, turn-taking, joint attention, and clarifying speech—without the need for scripted prompts. This approach offers a structured yet naturalistic context in which communication and collaboration can be observed, analyzed, and ultimately better understood. Both collaborative tasks were deployed in the same virtual environment, designed to maintain consistency while allowing specific design manipulations. In all scenarios, participants were placed in a virtual face-to-face interaction across a table (although they were physically separated in different rooms), each represented by a half-bodied avatar capable of head, eye, and hand movements. This spatial orientation—participants positioned opposite one another—was a deliberate design choice, as relative body placement in VR can significantly influence interaction dynamics. In reality, each participant is placed in a separate room. To support task understanding and reduce reliance on external instructions, each participant had access to an interactive virtual screen. These screens provided contextual information, including task objectives, rules, and example visuals. In addition, a red virtual button assigned per participant allowed them to signal task completion. Each task was capped at a maximum duration of 16 min, adding time pressure as an implicit coordination challenge. These interface elements remained consistent across both scenarios to ensure baseline comparability, while the interaction mechanics and information structures varied by task.

## 2. Method

### 2.1. Participants

The study involved 24 dyads (48 participants). Most dyads (21 out of 24) consisted of participants with a pre-existing relationship. Recruitment was structured to encourage sign-ups with a friend, ensuring familiarity between partners. This design choice was informed by prior research showing that partner familiarity can influence communication dynamics, such as non-verbal coordination, shared attentional focus, and gestural synchronicity [39,40,41,42]. Three dyads participated without prior familiarity. Participants were recruited primarily via the Facebook recruitment page of the Psychology Department at Ghent University, where calls for participation in paid experiments are regularly posted. All participants received financial compensation of EUR15 for their time and effort.

To ensure basic technological comfort, all participants were required to have experienced at least one prior VR session lasting a minimum of 30 min. This criterion was intended to reduce technical difficulties, limit the “wow effect” often observed in first-time VR users, and screen out individuals prone to motion sickness or discomfort in immersive environments. All participants were adults (18+), fluent in Dutch, and provided informed consent prior to participation. The study adhered to ethical standards and GDPR regulations enforced by Ghent University, ensuring anonymity during data collection and processing.

### 2.2. Apparatus

The experiment used the Meta Quest Prohead-mounted display (HMD) for its built-in extended reality (XR) tracking features, including eye tracking (90 Hz, 0.5° visual angle), facial muscle tracking (90 Hz), and camera-based hand tracking. These sensors allowed for high-resolution capture of head, eye, hand, and facial behaviors. The VR environment was developed in Unity and hosted using the Photon Engine, which enabled synchronized multi-user interaction and real-time voice communication. To reinforce the simulation of remote collaboration, participants were placed in separate rooms. Each space included a designated seat and a 2 × 2 m virtual boundary to support safe and natural interaction.

Avatar control and movement mapping were implemented through the Meta Horizon Interaction SDK. Standardized neutral avatars were used to ensure consistency and avoid bias. The SDK afforded the ability to use the tracking systems of the Quest Pro. Object tagging for gaze and gesture tracking was handled directly within Unity. Tags were applied to all interactive scene elements, enabling the system to log which objects were being observed or pointed at during the task. The virtual environment setup, including seating arrangement, dividers, and interface elements, is shown in Figure 1 and Figure 2 as a reference to the task layouts discussed in earlier sections. Facial tracking output consisted of 63 intensity values derived from the facial action coding system (FACS), capturing subtle expressions like eyebrow movement and cheek tension. These values were recorded in real time and applied to avatar expressions to support later analysis.

To clarify how participants’ real-world movements were captured and rendered as avatar behavior, Figure 3 illustrates the multi-user sensor-to-avatar pipeline used in this study. Each participant’s physical actions (eye, facial, head, and hand movements) were captured locally by the Meta Quest Pro’s onboard sensors. These raw signals were processed via the Meta SDK to extract gaze direction, facial expression parameters, and skeletal tracking data. Processed data were transmitted over a Photon PUN2 network session to the shared VR environment. Unity’s rendering engine then applied the incoming parameters to animate the partner’s avatar in real time, allowing both users to see each other’s movements from their own perspective. This bidirectional exchange occurred continuously for both users, enabling synchronized multi-user interaction in the shared scene.

### 2.3. Study Design and Procedure

The study followed a within-subjects design, with all dyads completing two collaborative VR tasks in a fixed order. Task 1 (Dynamic, Known Goal) was always presented before Task 2 (Structured, Unknown Goal). This sequence was chosen to guide participants from a more exploratory, spatially intuitive task toward one with structured, verbal coordination. Counterbalancing was avoided to prevent the structured communication patterns of Task 2 from bleeding into Task 1, where spontaneous collaboration was a key focus. A fixed order also ensured consistent comparisons of interaction behavior across conditions. Each task was time-limited to 16 min. Participants were instructed to complete the task as efficiently and collaboratively as possible. The total session—including both tasks and a short VR acclimatization period—was capped at 35 min. The fixed durations were designed to encourage timely interaction and decision-making without inducing artificial pressure. Throughout the experiment, an external observer view was available to the research team. This observer interface displayed the virtual scene from a top-down perspective of the shared table and included two additional camera feeds, each positioned at an oblique angle resembling an over-the-shoulder view of one participant facing the other. These camera views provided comprehensive visual coverage of both participants’ actions and interactions, allowing for real-time monitoring and supplementary qualitative analysis of collaborative behaviors. The cameras were initially configured using orthographic projection to ensure a consistent, undistorted field of view across sessions. Examples of these observer views can be seen in Figure 4 and Figure 5.

Before starting the VR tasks, participants completed a short intake procedure that included informed consent and a set of pre-questionnaires. These included the Big Five XS personality inventory and a 35-item self-assessment of communicative competencies, covering aspects such as collaboration tendencies, interaction comfort, flexibility, and persuasiveness. While these measures offer valuable context for understanding participant behavior, they are not the focus of the present study and are therefore not included in the analyses reported here.

An acclimatization phase preceded the experimental tasks. This brief period served a critical methodological purpose: to enhance participant comfort and reduce the likelihood of data contamination caused by unfamiliarity or disorientation in the virtual environment. Prior research has shown that gradual exposure to VR can mitigate symptoms of visually induced motion sickness (VIMS), such as nausea or dizziness, and improve the consistency of task performance [43]. Participants were given three minutes to familiarize themselves with interaction mechanics, avatar controls, and shared interface elements—including the completion button. They were encouraged to use the full acclimatization period but could proceed early if they felt ready.

#### 2.3.1. Task 1: Cube Assembly (Dynamic, Known Goal)

In this task, participants were instructed to collaboratively construct a 3D cube composed of six interlocking puzzle pieces. Completion required mutual effort, as the pieces were distributed in a way that prevented either participant from completing the structure alone. The task emphasized shared visual understanding and spontaneous coordination, and was to be completed as quickly as possible within a 16-min time frame. Participants worked across a central divider that partially obstructed their view of each other’s materials but maintained visual access to facial expressions and gestures. The divider doubled as the collaborative assembly platform, marking a clear spatial boundary between individual and shared work zones. Throughout the task, all puzzle pieces remained visible, including unused and misaligned ones, fostering continuous referencing and real-time mutual feedback.

Each participant received eight puzzle pieces, of which only three contributed to the final structure. Five were intentionally redundant to prompt discussion, misalignment, and clarification. Initially, participants could only manipulate their own pieces. Once a correct piece from one participant connected with a matching piece from the other, it became a shared component of the cube. These snapping events—five in total—marked visible progress toward completion. Task instructions and visual guidance were displayed on each participant’s interface screen. One page revealed a hint showing a correct piece. A second correct piece could be accessed by pressing a button named extra on the same interface. Each participant’s hints were different, though this was not disclosed to them. This ambiguity encouraged sharing, comparison, and subtle trust negotiation. The task was considered structurally complete once the cube was fully assembled, but formally ended only when both participants pressed their red completion buttons.

#### 2.3.2. Task 2: Structure Replication (Structured, Unknown Goal)

In this task, participants were asked to collaboratively build two identical flat geometric structures using a shared pool of 3D shapes. Working in alternating turns, each participant designed part of the structure and communicated it to their partner, who attempted to replicate the arrangement without visual access. The task required detailed verbal coordination and was to be completed within a 16-min time frame. The divider between participants was extended to fully block the view of each other’s workspace. In addition to this physical separation, all interactive elements—such as object movements and placed shapes—were rendered locally only, preventing any form of indirect visual access. Each participant could view and manipulate only their own pieces. As with Task 1, task instructions were accessible via the left-side screen, and a red button was provided for signaling formal completion.

Participants were given 34 3D shapes in various forms and colors—including circles, squares, T-shapes, L-shapes, and crosses. Despite their three-dimensional appearance, these objects were used to create a flat, tabletop construction rather than a volumetric form. While each participant had access to an identical set of shapes, the specific arrangement and placement of pieces were not identical by default and depended entirely on the active participant’s design choices. The task consisted of four turns in total—two per participant. Participants were free to decide who started. During each turn, the active participant selected and placed four shapes and then described the configuration verbally. The partner, listening in real time, attempted to recreate the layout on their side. Once both agreed that the replication was correct, the turn switched.

Each turn introduced an incremental step in the structure’s progression—at 4, 8, 12, and 16 placed shapes—mirroring the snapping milestones in Task 1. Participants had full freedom in what and how they placed, resulting in variable configurations and communication strategies. The task was considered complete once both participants verbally confirmed that their structures matched. As in Task 1, the final step involved pressing the red button to formally signal completion.

#### 2.3.3. Comparison of Task 1 and Task 2

The two collaborative VR tasks were deliberately designed to elicit contrasting communication strategies by manipulating access to visual information, structuring of interaction, and control over task progression. Task 1 encouraged dynamic, parallel coordination around a visible and continuously accessible goal. The cube’s partial assembly on a shared platform allowed participants to jointly observe progress, adjust actions in real time, and communicate through gestures, gaze, and minimal verbal clarification. The T-shaped divider maintained individual workspaces while still enabling facial and gestural visibility, reinforcing non-verbal interaction as the dominant coordination channel.

In contrast, Task 2 removed access to the partner’s environment entirely. With both physical visibility and digital rendering of partner actions blocked, collaboration was dependent on verbal communication and turn-taking. The construction goal was not externally defined or shown; instead, it emerged incrementally through the participants’ shared efforts. Verbal descriptions became the primary tool for conveying spatial relationships, sequencing, and correction. Rather than spontaneous joint action, coordination followed a tightly structured sequence, with four fixed turn switches and mutual confirmations marking task progression.

The emphasis in Task 1 thus lay in real-time gestural collaboration and shared attention on a known target, while Task 2 required careful verbal negotiation, perspective-taking, and sequential planning to reach alignment on an unknown and evolving structure. These differences reflect two distinct approaches to eliciting soft skill expressions: one driven by co-presence and emergent joint action, the other by structured communication and mutual understanding without visual support. By comparing behavior across these two setups, we examine how scenario design influences the form, timing, and modality of collaborative interactions in multi-user immersive environments.

### 2.4. Analysis Methodology

To evaluate how task design influenced dyad-level interaction behaviors, we extracted a set of quantitative metrics from multiple data streams, including XR-based signals (eye, hand, and facial tracking), voice recordings, and logged interaction events. These metrics are described in detail in the following subsections and capture different aspects of collaborative behavior—for example, durations of specific behaviors, frequencies of events, proportions of time spent on particular activities, and counts of interaction episodes.

Where relevant, time-based metrics were computed as a ratio relative to total task duration, allowing comparisons across tasks of equal length. The following formula was used for calculating such ratios:(1)$$\mathrm{Ratio}=\left(\frac{\mathrm{Seconds}\;\mathrm{of}\;\mathrm{observed}\;\mathrm{behavior}}{\mathrm{Total}\;\mathrm{task}\;\mathrm{time}}\right)\times 100$$

For dyadic analyses, values from both participants were averaged to produce a team-level metric:(2)$$\mathrm{Team}\;\mathrm{Ratio}=\frac{\mathrm{Player}\;1\;\mathrm{Ratio}+\mathrm{Player}\;2\;\mathrm{Ratio}}{2}$$

These formulas follow conventions in multimodal and social interaction research, where behavior normalization and averaging are commonly used to enable comparability across sessions and reduce participant-level variability. Similar strategies have been employed in studies on social signal processing and dyadic collaboration [44,45]. To statistically compare behaviors between Task 1 and Task 2, we used paired-sample *t*-tests. Bonferroni correction was applied to control for multiple comparisons, and Cohen’s d was reported to indicate the magnitude of observed differences.

Beyond direct statistical comparisons, we also investigated whether or not patterns of non-verbal and paraverbal behavior could be used to distinguish between task conditions in an unsupervised manner. This analysis aimed to assess whether or not dyads’ interaction profiles naturally clustered according to task type, without relying on predefined task labels.

As a first step, we temporarily treated the task number as a label and trained a Random Forest classifier on the full set of computed behavioral metrics. This step was used solely for identifying which features carried the highest importance for distinguishing between the two tasks. Subsequently, we selected a subset of the most informative features, based on this ranking, for input into the clustering analysis. KMeans clustering was then performed on the selected features. To determine the optimal number of clusters, we calculated silhouette scores across a range of possible cluster counts. This methodological pipeline allowed us to explore whether or not the tasks could be differentiated purely from behavioral data patterns.

The following sections provide detailed descriptions of each type of behavioral metric analyzed in this study, including how they were defined, measured, and computed from the collected data streams.

#### 2.4.1. Task Completion Metrics

Task performance was assessed using two key indicators. **Completion Time** referred to the total number of seconds from task initiation to the moment both participants pressed their virtual red buttons, signaling mutual agreement on task completion. **Completion Rate** was defined differently across the two tasks. For Task 1, it was measured based on the number of successful snap events—each representing the correct connection of two puzzle pieces—with a total of six possible snaps. For Task 2, completion rate reflected both structural accuracy and progression. Specifically, we multiplied the replication accuracy (i.e., the percentage similarity between the two constructions) by the number of pieces placed, and then divided this value by the task target of 16 shapes. This approach captures both how much of the structure was built and how precisely it matched the intended result.

#### 2.4.2. Gaze-Based Metrics

Eye tracking data were collected at two sample rates −90 Hz for the Quest Pro’s default sample rate and resampled to 30 Hz to align with Unity’s frame-based logging. Collider tagging enabled gaze-object mapping. We derived the following measures:**Task Object Focus:** Proportion of time spent looking at task-relevant elements (e.g., puzzle pieces or geometric shapes).**Eye Contact:** Defined as moments where both participants simultaneously looked at each other’s faces, allowing for a ±1 s validation window.**Joint Attention:** Instances where both participants looked at the same object concurrently. In Task 2, where participants could not see each other’s workspace, this was inferred based on simultaneous gaze directed at identical but separate objects.**One-Way Gaze:** Occurrences where one participant looked at their partner while the other focused elsewhere, indicating potential turn initiation or attention-seeking behavior.

Gaze proportions were calculated relative to total task duration and collectively represent 100% of visual attention. These were categorized into three target domains: task-related elements (e.g., puzzle pieces, geometric shapes), partner-related elements (e.g., the other participant’s avatar face or body), and environment-related elements (e.g., background objects or empty space). Task object focuscaptured time spent looking at task-relevant elements, and joint attention was treated as a sub-component of this category, denoting periods where both participants looked at the same object simultaneously. In contrast, eye contact and one-way gaze were categorized as partner-directed behaviors, reflecting direct and asymmetrical gaze engagement, respectively.

#### 2.4.3. Hand Gesture Metrics

Hand tracking was event-based, logging the start and end times of recognized gestures. Pose recognition was implemented via Boolean logic on finger positions using built-in headset cameras. The Meta SDK supports several predefined gestures (e.g., thumbs-up, stop), and a custom pointing pose was added. This pointing gesture also allowed for tagging which object was being referenced at each detection instance.

**Grasping Behavior:** Logged whenever participants successfully grabbed or attempted to grab an object, indicating task engagement.**Expressive Gestures:** Calculated as the proportion of total task time during which predefined gestures (e.g., thumbs-up, stop) were actively detected. Rather than counting individual gesture instances, this metric reflects the cumulative duration of time that each gesture was held. Each gesture was tracked individually.

#### 2.4.4. Facial Expression Metrics

Facial activity was sampled at 90 Hz and analyzed using 63 blendshape values provided by the Meta Quest Pro. These blendshapes are mapped to facial action coding system (FACS) action units (AUs), and each AU is represented as a continuous value ranging from 0 to 1. As previously mentioned, detected facial expressions are defined by combinations of AUs. To reliably assess facial activity while minimizing the influence of noise, it was necessary to establish thresholds for interpreting AU intensity. While our data remained continuous, we referenced the DISFA dataset [46], which annotates AU intensities on a discrete 0–5 scale, to provide qualitative context for interpreting the meaning of different intensity levels. Table 1 illustrates how we conceptually mapped the continuous Meta Quest Pro values onto these qualitative labels for interpretative purposes only.

For this study, we focused on **non-neutral expressions**, defined as moments in which two or more AUs exceeded a specified intensity threshold simultaneously. To systematically explore sensitivity, we tested thresholds ranging from 0.1 to 0.5, corresponding to the lower to mid-levels of our conceptual mapping between the continuous 0–1 scale and the qualitative DISFA intensity labels. Specifically, these thresholds span from “trace” activation to approximately the midpoint of “marked or pronounced” intensity.

Expression changes were detected by examining sliding windows of 3 s, during which we calculated the absolute differences in AU values compared to the preceding window. A significant non-neutral expression was registered when at least two AUs increased above the chosen threshold within the same time frame. This approach avoids reliance on predefined categorical emotion labels and instead captures the presence and magnitude of facial expressiveness across tasks while maintaining the underlying continuous nature of the data.

#### 2.4.5. Voice Activity Metrics

We quantified speech behavior using three complementary voice activity detection (VAD) tools: Parselmouth [47], Librosa [48], and WebRTC VAD [49]. These methods differ in detection granularity and noise tolerance: Parselmouth provides precise phonation boundaries via the Praat backend, Librosa applies energy-based segmentation tuned for speech/music signals, and WebRTC VAD uses a real-time frame-based algorithm optimized for low-latency speech detection. All VAD analyses used consistent parameters across tasks and participants. For Librosa, we applied a threshold of 25 dB below peak energy to segment speech activity. WebRTC VAD was set to aggressiveness mode 2 with a 30 ms frame window. Parselmouth used an intensity threshold of 25 dB. Across all tools, segments shorter than 0.1 s were excluded during post-processing to reduce false positives from brief noise or micro-pauses.

For each dyad and task, we aggregated the outputs from all three tools by computing the mean value for each variable. This ensemble approach improves robustness by minimizing bias introduced by any single method and accommodates variability in speaking styles and recording conditions.

**Number of Speech Segments:** Total count of uninterrupted speaking episodes per dyad.**Segment Duration:** Minimum, maximum, and average length (in seconds) of speech episodes, reflecting rhythm and pacing.**Speaking Percentage:** Proportion of total task time during which speech was detected.

In addition to basic speech segmentation, we analyzed overlapping speech to capture simultaneous verbal contributions—a key indicator of conversational dynamics and interactivity. Overlapping speech was defined as instances where both participants’ speech segments overlapped in time by at least 30 ms. For each dyad and task, we computed the total number of overlapping speech segments, the cumulative duration of overlap (in seconds), and the proportion of overlapping speech relative to the total speaking time. These overlapping metrics were derived from the aggregated ensemble VAD output, ensuring consistency across detection methods.

### 2.5. Study Hypotheses

While communication is inherently multimodal—with verbal content playing a central role—this study focuses on non-verbal and paraverbal markers. For now, we concentrate on interaction dynamics that unfold through observable timing, gaze, gesture, and vocal delivery patterns. To guide this inquiry, we formulate hypotheses based on the expected effects of task structure on behavior. For clarity, we refer to the two conditions as follows: Task 1 represents the Dynamic, Known Goal condition—featuring spontaneous, parallel collaboration around a visible objective. Task 2 represents the Structured, Unknown Goal condition—requiring sequential, turn-based coordination and the co-construction of a concealed structure.

We begin with hypotheses concerning gaze and gestural behavior. In Task 1, participants are expected to allocate more attention to the visible task elements, while in Task 2 we anticipate more partner-directed gaze, including increased instances of eye contact, due to the absence of a shared external referent (H1). This expectation is grounded in studies showing that shared visual access reduces the need for gaze-based coordination, whereas in its absence people rely more heavily on mutual gaze and eye contact to manage interaction and establish shared attention [50,51]. We also expect differences in how joint visual focus manifests. In Task 1, both participants operate within the same visual space, enabling direct co-attention on individual objects. In Task 2, visual co-attention must be inferred—participants see only their own identical object set and rely on synchronized descriptions to match attention implicitly. As such, we hypothesize that moments of inferred shared focus will occur more frequently in Task 2, where the task design encourages aligned visual reference through speech and structure building (H2). This aligns with work on joint attention showing that coordination becomes more effortful and behaviorally rich when mutual referents must be inferred rather than observed [50].

In terms of physical expressiveness, we hypothesize that Task 1 will elicit a higher frequency of gestures due to its unstructured pacing and need for constant spatial adjustment (H3). Spontaneous gestures often accompany speech in tasks involving spatial reasoning or physical manipulation, especially when interaction is fast-paced and externally anchored [52]. Conversely, Task 2 may rely more heavily on facial cues and expressions, as the structured pauses, explicit role switches, and lack of shared objects allow participants more opportunity to engage expressively through facial feedback (H4). Even outside VR, facial feedback has been shown to play a key role in regulating social interaction—providing backchannel cues and aiding turn negotiation when explicit object references are unavailable [53].

Turning to paraverbal patterns, we expect that Task 2 will involve longer uninterrupted speech episodes, as participants deliver entire segments of instruction before their partner responds (H5). This reflects the structured, one-at-a-time nature of the task, which favors extended speaker turns. In contrast, the fast-paced and shared-object nature of Task 1 is likely to generate more overlapping speech as participants coordinate in real time (H6). Additionally, we hypothesize that speech in Task 1 will be more fragmented and interactive—reflected in a higher number of distinct speech segments—while Task 2 will encourage slower, more deliberate phrasing (H7). These hypotheses are supported by turn-taking research showing that spontaneous conversations involve frequent short turns, overlaps, and rapid back-and-forth timing, while more formal or instruction-driven exchanges result in longer, more coherent segments [54,55].

Together, these hypotheses reflect our theoretical stance that task design actively shapes behavioral emergence in VR. Rather than treating communication as a static capacity, we view it as an adaptive response to environmental constraints and affordances—making scenario structure a critical determinant of how soft skills are expressed and observed. To clarify how our analyses address the theoretical expectations set forth in the study design, Table 2 summarizes the mapping between each hypothesis and the associated behavioral metrics.

## 3. Results

Due to technical issues affecting data integrity, the analyses presented in the following sections are based on 23 dyads, out of the 24 dyads that participated. Group 16 was excluded due to incomplete data logs for Task 2, likely caused by network-related synchronization issues. Although audio recordings for this dyad were preserved, the absence of complete multi-modal data prevents their inclusion in a comparative analysis across both tasks. All participants successfully completed the experimental tasks. For all subsequent visualizations, numeric labels next to data points indicate how many dyads share identical values, aiding interpretation where overlap occurs.

In the following visualizations, an asterisk * indicates p<0.05; ** indicates p<0.01.

### 3.1. Task Completion Analysis

Each task had a fixed maximum duration of 16 min. Participants were instructed to complete the tasks as fast as possible within this time limit. Despite the time constraint, several dyads finished early, with the shortest observed durations being 4 min 53 s for Task 1 and 7 min 44 s for Task 2.

Figure 6a shows the completion time of each dyad. Seven dyads reached the 16-min time cap in Task 1, compared to only five in Task 2. No significant difference in overall task duration was observed between Task 1 (M=12.21 min, SD=4.20 min; ≈ 12 min 13 s) and Task 2 (M=13.65 min, SD=2.68 min; ≈ 13 min 39 s), t(22)=1.40, p=0.1764, Bonferroni-corrected p=1.000, d=0.29.

Figure 6b shows completion rates across tasks. Task 1 yielded an average completion rate of M=67.82%, SD=34.20, whereas Task 2 averaged M=94.22%, SD=12.15. Two dyads failed to assemble any correct pieces in Task 1, resulting in a 0% completion score. In contrast, no dyad in Task 2 scored below 50%. A paired-samples *t*-test confirmed a significant difference in completion rates between tasks, t(21)=3.50, p=0.0021, which remained significant after Bonferroni correction (p=0.015), d=0.75. This indicates that task success was significantly higher in Task 2.

### 3.2. Gaze Behavior Analyses

To evaluate how visual attention was distributed and coordinated within dyads, we analyzed three gaze-related metrics: task object focus, joint attention, and eye contact. Together, these features reflect how participants navigated shared attention and interpersonal engagement during collaboration. To complement the existing variables, we also included joint attention frequency as an additional metric. While not explicitly outlined in the methodological section, it offers valuable insight into the temporal dynamics of shared attention, capturing how often alignment occurred rather than just how long it was maintained.

Figure 7a shows the average proportion of partner-directed gaze across both tasks. In Task 1, dyads displayed an average of 2.48% partner-directed gaze (SD=2.16), corresponding to 18.17 ± 15.82 s of eye contact during a mean task duration of 12 min and 13 s. In Task 2, the average dropped to 1.46% (SD=1.37), equivalent to 11.96 ± 11.16 s, with a slightly longer mean duration of 13 min and 39 s. While a paired-samples *t*-test indicated a nominally significant difference between conditions (t(22)=−2.08, p=0.0497), this effect did not survive Bonferroni correction (p=0.9438). Taken together, eye contact accounted for only a small fraction of the interaction in both tasks—approximately 12–18 s across 12–14 min—and did not differ robustly after correction. Variability in eye contact was also greater in Task 1, indicating more between-dyad heterogeneity in partner-directed gaze under dynamic, parallel collaboration.

Figure 7b displays the proportion of gaze directed at task-relevant elements. In Task 1, participants spent on average M=64.56% (SD=9.76) of their viewing time on task objects, corresponding to approximately 7 min 53 s (±1 min 11 s) of the average duration. In Task 2, this increased to M=78.90% (SD=5.51), or roughly 10 min 46 s (±45 s) of the average task duration. This difference was highly significant, t(22)=8.83, p<0.0001, and remained so after Bonferroni correction (p<0.0001), with a large effect size (d=1.84). The higher proportion in Task 2 suggests that its structured, turn-based coordination anchored participants’ visual attention more consistently to relevant objects, whereas Task 1’s dynamic pacing and shared visual field allowed more gaze to drift toward non-task elements.

Joint attention duration is illustrated in Figure 7c. In Task 1, participants spent on average M=20.13% (SD=10.95) of the trial in joint attention, corresponding to approximately 2 min 27 s (±1 min 20s) of the average task duration. In Task 2, this increased to M=38.07% (SD=8.59), or roughly 5 min 12 s (±1 min 10s) of the average task duration. This difference was statistically significant, t(22)=9.00, p<0.0001, and remained so after Bonferroni correction (p<0.0001), with a strong effect size (d=1.88). The marked increase in Task 2 aligns with its design constraints, which required frequent coordinated reference to the same conceptual target despite participants not sharing a physical view of each other’s objects.

Figure 7d presents joint attention frequency. In Task 1, dyads aligned their visual focus an average of M=69.57 times (SD=38.65) during the 12 min 13 s average task duration. In Task 2, this increased to M=105.43 episodes (SD=28.27) over the 13 min 39 s average task duration. The difference was statistically significant, t(22)=3.62, p=0.0015, and remained significant after Bonferroni correction (p=0.0287), corresponding to a medium-to-large effect size (d=0.76). The higher frequency in Task 2 reflects the need for continual verification of shared focus when no visible common workspace is available, requiring more frequent gaze-based coordination.

### 3.3. Hand Gesture Analyses

Figure 8a shows the overall proportion of time spent performing expressive gestures across tasks. Dyads gestured slightly more in Task 1 (M=27.06, SD=10.84) than in Task 2 (M=23.95, SD=10.63). However, this difference was not statistically significant, t(22)=−1.64, p=0.1151, and remained non-significant after Bonferroni correction (p=1.000), with a small-to-moderate effect size (d=−0.34).

While several predefined gestures were logged individually—including thumbs-up, stop, and others—their occurrence was too infrequent across both tasks to support statistical analysis. Consequently, only the pointing gesture was selected for more detailed examination due to its higher frequency and its functional role in task-related referencing. Gesture-specific summary statistics are provided in Table 3, illustrating the sparsity of non-pointing gestures under both task conditions. As shown in Figure 8b, pointing gestures occurred significantly more often in Task 1 (M=9.85, SD=5.80) than in Task 2 (M=7.05, SD=4.54). This difference was statistically significant, t(22)=−3.62, p=0.0015, and remained significant after Bonferroni correction (p=0.029), indicating a medium-to-large effect size (d=−0.75).

### 3.4. Facial Expression Analysis

Facial expression activity was derived from real-time blendshape tracking, providing continuous intensity values between 0 and 1 for 63 Facial Action Units (AUs). As described in the methodology, five thresholds ranging from 0.1 to 0.5 were evaluated to assess sensitivity to expressive changes. For this analysis, results are presented using a 0.3 threshold—chosen for its balance between noise suppression and expressivity detection, based on AU scaling practices from the DISFA dataset.

Figure 9 illustrates the ratio of non-neutral facial expressions observed across both tasks. On average, participants exhibited a higher proportion of non-neutral expressions in Task 1 (M=11.94, SD=7.16) than in Task 2 (M=7.09, SD=4.10). A paired-samples *t*-test confirmed a statistically significant decrease between tasks (t(22)=−3.66, p=0.0014), which remained significant after Bonferroni correction (p=0.0264), reflecting a medium-to-large effect size (d=−0.76). A majority of dyads (18 out of 24) demonstrated greater facial expressivity in Task 1.

### 3.5. Voice Activity Analyses

Figure 10 summarizes the voice activity metrics across both tasks. Participants produced significantly longer speech segments in Task 2 (M=0.79, SD=0.12 s) than in Task 1 (M=0.70, SD=0.10 s), t(21)=4.26, p<0.001, Bonferroni-corrected p=0.004, d=0.87, indicating more continuous verbal contributions during the structured task. The number of speech segments was significantly higher in Task 1 (M=32.41, SD=4.96) than in Task 2 (M=28.33, SD=4.31), t(21)=−5.12, p<0.0001, Bonferroni-corrected p=0.001, d=−1.05, reflecting a more fragmented and interactive conversational style in the dynamic collaborative task. In contrast, the percentage of speaking time was slightly higher in Task 2 (M=37.18%, SD=4.21%) than in Task 1 (M=35.64%, SD=4.09%). However, this difference was not statistically significant, t(21)=1.52, p=0.141, and remained non-significant after Bonferroni correction (p=1.000), with a small effect size (d=0.31).

Regarding overlapping speech (Figure 11), no significant differences were observed between tasks in either the total duration of overlap or the number of overlapping segments. The total duration of overlapping speech averaged 10.14 s (SD=3.58) in Task 1 and 9.67 s (SD=3.29) in Task 2, t(21)=−0.81, p=0.428, Bonferroni-corrected p=1.000, d=−0.16. Similarly, the number of overlapping speech segments did not differ significantly across tasks.

These results suggest that while Task 2 promoted longer, more continuous speech contributions, Task 1 elicited shorter, more frequent verbal exchanges consistent with its dynamic and co-active structure.

### 3.6. Feature Relationships

To evaluate whether or not participants’ behavioral profiles could distinguish the two collaborative tasks in an unsupervised setting, we performed a clustering analysis using the most informative features from the earlier Random Forest importance ranking. From this ranking, the three highest-ranked features—joint attention, task focus, and average speech segment duration—were selected as input variables for clustering. Silhouette scores were computed for solutions ranging from k=2 to k=10. The highest silhouette score was observed at k=2, with a value of 0.3969, indicating a moderate clustering structure. Using k=2, KMeans clustering was performed on the selected features. The resulting clusters are visualized in Figure 12, depicting how dyads distributed across the feature space.

To evaluate the alignment between clusters and actual task conditions, each cluster was mapped to the most frequent task label within it. The confusion matrix for this mapping is shown in Table 4.

This cluster-to-task mapping yielded a macro-averaged F1 score of 0.8849, suggesting that the top three behavioral features captured meaningful differences between the two task conditions, even in an unsupervised setting.

## 4. Discussion

This study investigated how task constraints in virtual reality influence multimodal collaboration dynamics, with a focus on non-verbal and paraverbal behavior. By contrasting a spatially co-present, visually shared task (Task 1) with a structured, visually restricted one (Task 2), we explored how scenario design actively shapes how dyads coordinate, attend, and express themselves in immersive environments.

Overall, our results highlight that collaborative behavior in VR is highly dynamic and context-dependent, adapting continuously in response to task affordances and limitations. While we expected Task 2 to promote more partner-directed gaze due to the lack of a shared visual reference (H1a), our data did not support this. Eye contact and one-way gaze frequencies did not differ significantly between tasks, contrary to H1a. Instead, participants in Task 2 allocated significantly more visual attention to their own task elements. This finding contradicts H1b, which predicted higher task-focused gaze in Task 1 due to its open, shared visual environment. This likely reflects the increased cognitive and manual demands imposed by the structured, turn-based activity, which may have consumed attentional resources and anchored gaze locally rather than toward the partner. According to cognitive load theory, higher perceptual and working memory demands reduce the capacity to process task-irrelevant stimuli, leading individuals to prioritize local task information over social or environmental cues [56,57]. Despite this local anchoring of attention, joint attention—measured both in duration and frequency—was significantly higher in Task 2, supporting H2 and indicating that participants still succeeded in coordinating their focus through verbal scaffolding. This suggests that while direct visual cues were unavailable, participants effectively used verbal descriptions to create moments of inferred shared attention.

In terms of non-verbal expressivity, gesture analysis revealed a higher frequency of pointing behavior in Task 1, confirming H3, where spatial referencing and object manipulation occurred in real time. Although the system captured multiple predefined gestures (e.g., thumbs-up, stop), their occurrence was too sparse across both tasks to yield meaningful comparisons, suggesting that more deliberate emblematic gestures were less relevant in this collaborative context. Instead, pointing emerged as the dominant gestural modality, serving as an immediate and intuitive means to direct attention and coordinate shared actions. Although we hypothesized that structured turn-taking in Task 2 might increase facial feedback (H4), we observed the opposite: facial expressivity was significantly greater in Task 1, contradicting H4. This may be explained by the more fluid and reactive nature of Task 1, which allowed for frequent, brief facial signals—smiles, raised brows, or other micro-expressions—as participants negotiated alignment in a shared space. In contrast, Task 2 may have introduced a form of expressive inhibition, as the complexity of object placement and pacing of turn-based construction demanded sustained concentration and reduced opportunities for spontaneous affective signaling. This pattern aligns with cognitive load theory, suggesting that high task demands may suppress spontaneous facial expressivity as attentional resources are redirected toward task-specific processing [56,57]. Facial expressions are strong indicators of cognitive and emotional states and often outweigh auditory cues in signaling uncertainty or effort [58]. Under higher cognitive load, as in Task 2, producing facial signals may compete with the mental effort needed for precise verbal communication. Thus, the reduced facial expressivity we observed likely reflects an adaptive shift of cognitive resources from non-verbal cues toward task execution and speech, consistent with Dijkstra et al.’s findings.

With regard to paraverbal behavior, our findings indicate that Task 2 prompted longer average speech segments and fewer overall speech segments compared to Task 1, supporting H5 and H7, respectively. This suggests that the structured, turn-based nature of Task 2 encouraged participants to produce longer, uninterrupted verbal contributions, consistent with expectations based on the task design. In contrast, Task 1 was characterized by shorter, more reactive speech episodes, aligning with its dynamic, co-active nature where rapid exchanges and immediate feedback were required to coordinate joint actions. Interestingly, despite Task 2’s more sequential structure, overlapping speech was not significantly reduced compared to Task 1, failing to support H6. This finding may reflect the inherent challenges of maintaining strict conversational boundaries in collaborative problem-solving, even in turn-based tasks [59]. Moreover, the percentage of speaking time did not differ significantly between tasks, suggesting that while speech rhythm and segmentation patterns shifted, overall verbal engagement remained robust across conditions. These results highlight the adaptive flexibility of verbal communication in VR collaboration, where participants adjust their speech patterns to compensate for changes in visual access and task structure.

Beyond direct comparisons of individual behavioral metrics, our clustering analysis demonstrated that joint attention, task focus, and average speech segment duration together provided a strong basis for distinguishing between the two task conditions without prior task labels. The clear separation of dyads into two clusters, as evidenced by a macro-averaged F1 score of 0.88, underscores that the interaction patterns elicited by each task were not merely subtle statistical trends but reflected robust, distinguishable behavioral profiles. This finding suggests that task constraints in VR generate characteristic multimodal signatures—combinations of gaze, speech, and attention metrics—that can potentially serve as reliable indicators of collaborative style or task engagement. Such unsupervised analyses open promising avenues for developing automatic classifiers that detect task context or assess collaborative dynamics in real time, enabling more adaptive VR systems and personalized feedback mechanisms for training and assessment.

A critical, unintended influence on behavior stemmed from interaction mechanics. Participants frequently experienced frustration manipulating small objects, particularly in Task 2. The lack of haptic feedback led to coordination breakdowns and redirected attention away from joint planning toward solo problem-solving. These interactional frictions were not trivial: they interfered with natural expressivity and required participants to invest effort in compensating for mechanical shortcomings. Interestingly, despite these challenges, verbal coordination remained relatively robust in Task 2, suggesting that when non-verbal channels are constrained—either by design or by technical limitations—participants can adapt by strengthening verbal channels. This behavioral compensation underscores the resilience of human communicative adaptability but also highlights the importance of improving interaction fidelity in VR systems. Beyond these findings, the study surfaced emergent behaviors unique to the virtual context. Some participants exploited the affordances of the medium—such as walking through virtual tables or adjusting their avatar’s viewpoint unrealistically—to gain a better perspective. These actions, while impossible in physical settings, were used strategically and highlight how VR may elicit novel forms of collaboration that diverge from real-world norms. Whether such behaviors should be constrained or embraced depends on the goals of the VR experience: training environments may prioritize fidelity, while exploratory or educational settings may benefit from such flexibility.

These findings underscore the necessity for future research to examine how VR scenario design can systematically influence collaborative behavior, both at group and individual levels. A critical avenue is the development of robust behavioral taxonomies and rating systems that integrate verbal, non-verbal, and paraverbal channels. Such frameworks could allow researchers and practitioners to map specific patterns of interaction—such as joint attention, balanced turn-taking, or expressive gestures—to desirable soft skills outcomes. Equally important is establishing connections between sensor-derived metrics and human evaluative judgments, including expert observations and participant self-assessments. Bridging these perspectives would enable validation of objective measures and support the creation of scalable tools for training and assessment. Ultimately, integrating multimodal behavioral data with human interpretive insights holds significant potential for designing adaptive VR systems capable of delivering real-time feedback and personalized interventions grounded in empirically defined markers of effective collaboration.

### Limitations and Future Directions

Several limitations of our study warrant consideration. First, although our analyses focused on non-verbal and paraverbal behavior, these interactions were mediated through simplified avatars. The fidelity of eye-gaze, facial expression, and hand tracking in current VR systems remains limited, which can constrain how well partners interpret each other’s intentions. Prior work in social VR has shown that greater avatar realism and more expressive non-verbal channels can enhance co-presence and the interpretability of social cues, thereby influencing interaction outcomes [60,61,62]. Our findings should therefore be interpreted as indicative of relative differences between task conditions rather than as direct proxies for real-world behavior.

Second, our sample comprised predominantly familiar dyads. Familiarity may facilitate coordination and reduce communicative effort, limiting the generalization of our findings to strangers or mixed-expertise dyads. Although this recruitment bias is noted in the Participant section, future work should systematically compare familiar and unfamiliar dyads to assess how relationship dynamics influence non-verbal and paraverbal behavior.

As previously stated, we observed technical constraints related to hand tracking. Because participants relied on optical hand tracking to grasp and place virtual objects, imprecise tracking occasionally led to frustration and disrupted task flow. In the post-session debriefing, 31 participants reported more difficulty manipulating objects in Task 2 (Structured, Unknown Goal), where precise placement was required. We recommend that future studies use controller-based manipulation or improved tracking systems when object handling is central to the task. We did not include formal measures of user comfort or quality of experience. While participants informally reported feeling more strain during precise object manipulation, future experiments should incorporate validated subjective measures (e.g., presence, workload, frustration) to better understand how VR task design influences both experience and the interpretation of partner behaviors.

Finally, our study was limited to two tasks with specific structural properties (dynamic versus structured, known versus unknown goal) and a sample size of 23 dyads, which was adequate for the repeated-measures comparisons at the dyad level reported here. However, human interaction is inherently complex and multimodal, and future work should incorporate a broader range of tasks varying in complexity, role symmetry, and feedback modality, alongside substantially larger datasets. Such expansions would allow for robust testing of the generalizability of our findings and enable multilevel analyses that capture both individual- and dyad-level variation, providing a more nuanced understanding of how behavioral patterns emerge and differ across contexts.

## 5. Conclusions

Our findings demonstrate that behavioral expression in VR is profoundly shaped by scenario design. Non-verbal coordination flourishes in shared visual spaces, where spontaneous gestures and joint attention drive collaboration. Conversely, verbal strategies dominate when visual access is limited, but only when tasks are structured to support sequential alignment. These results underscore that communication strategies are not merely reflections of individual skill but are actively constructed in response to the social, spatial, and technical conditions of the environment. By combining detailed multimodal behavioral metrics with task-driven design, this study advances our understanding of how small shifts in VR scenarios produce meaningful differences in collaborative behavior. While not all hypotheses were confirmed, the observed adaptations highlight the flexibility of human communication in immersive contexts. Our findings contribute to a growing body of research advocating for more ecologically valid, data-driven assessment tools [9,23,41]. Virtual reality offers a powerful platform for this purpose, enabling researchers and practitioners to observe behavior in context and manipulate conditions under which it emerges. This opens promising pathways for scalable, objective approaches to soft skills training and assessment—approaches grounded in observable behavior rather than in retrospective judgment or self-report. As immersive technologies mature, designing with behavioral intent becomes not just an option but a necessity.

## Figures and Tables

**Figure 1 sensors-25-05498-f001:**
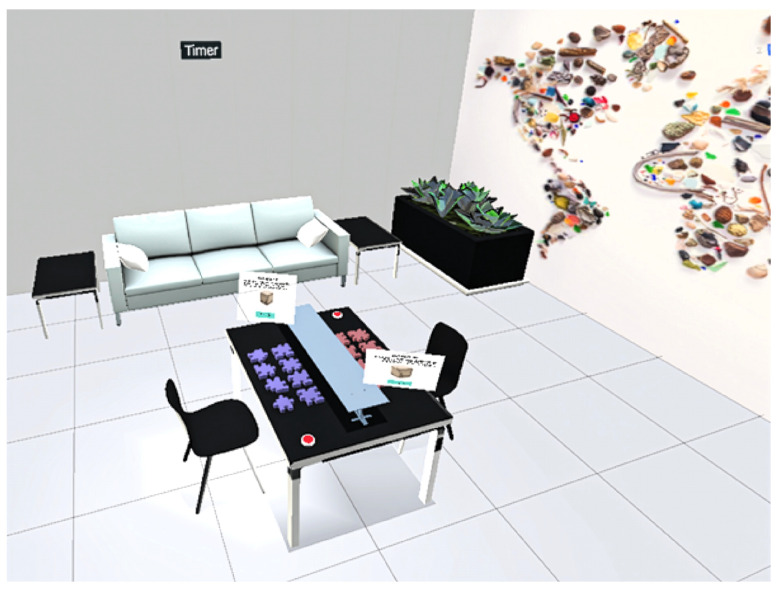
Task 1: Cube assembly room setup.

**Figure 2 sensors-25-05498-f002:**
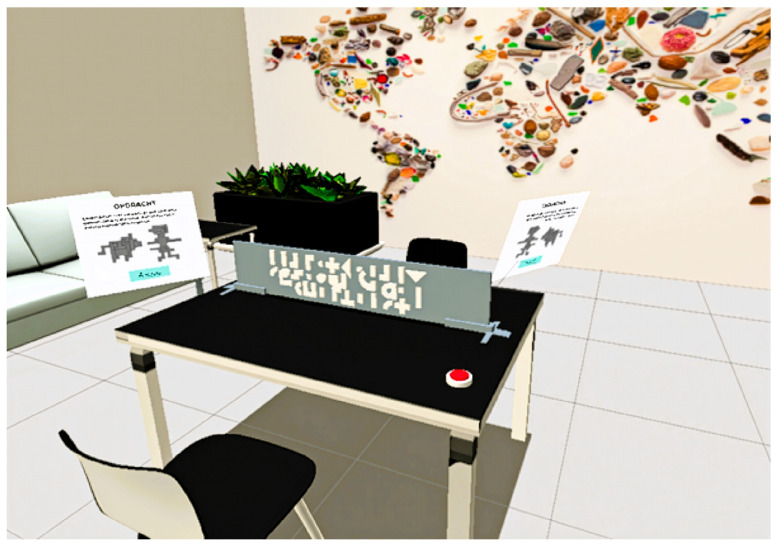
Task 2:Structure replication room setup.

**Figure 3 sensors-25-05498-f003:**
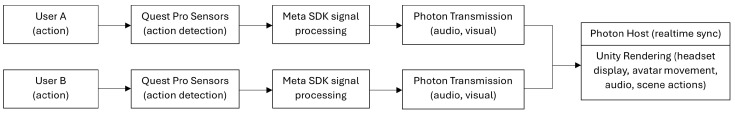
Multi-user interaction and processing flow in the VR system. Actions from each user are detected by Quest Pro sensors, processed via the Meta SDK, transmitted via Photon for real-time synchronization, and rendered in Unity for both users.

**Figure 4 sensors-25-05498-f004:**
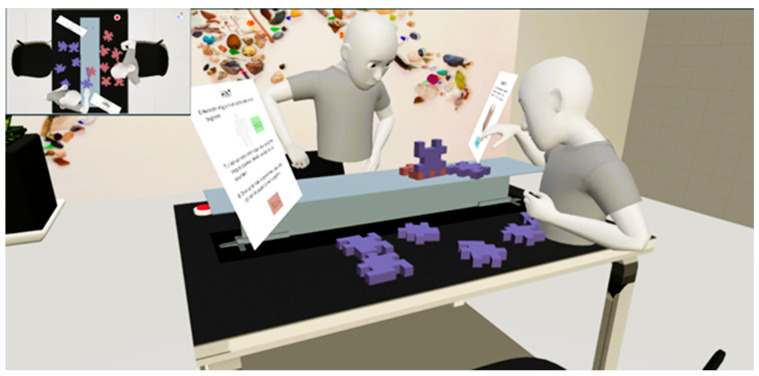
Task 1: Cube assembly dyad group interaction (Observer view).

**Figure 5 sensors-25-05498-f005:**
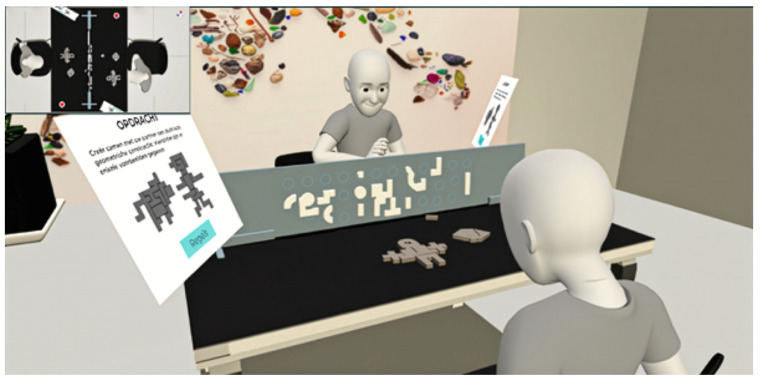
Task 2: Structure replication dyad group interaction (Observer view).

**Figure 6 sensors-25-05498-f006:**
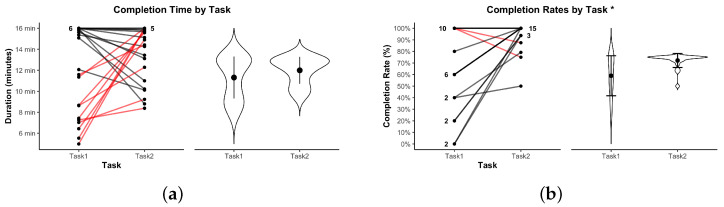
Completion metrics by task. (**a**) Completion time; red lines indicate dyads that took longer to complete Task 2 compared to Task 1. (**b**) Completion rate; red lines indicate dyads with a lower completion rate in Task 2 compared to Task 1. Violin plots show distributions. * indicates p<0.05.

**Figure 7 sensors-25-05498-f007:**
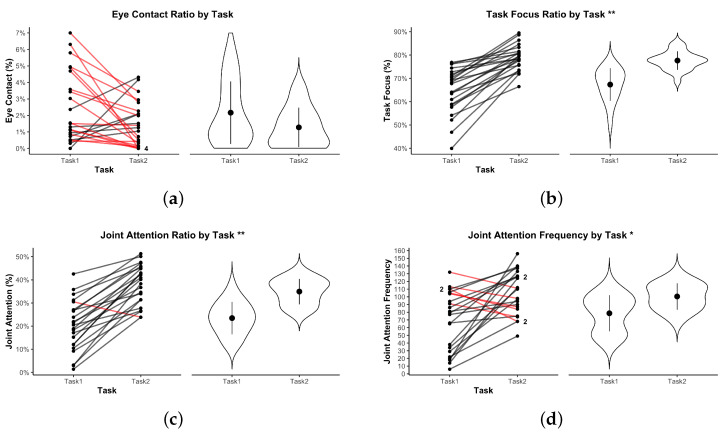
Eye-gaze metrics for Task 1 vs. Task 2. (**a**) Eye contact ratio. (**b**) Task object focus. (**c**) Joint attention duration. (**d**) Joint attention frequency. Violin plots show distributions. Red lines indicate dyads where the metric decreased from Task 1 to Task 2. * indicates p<0.05; ** indicates p<0.01.

**Figure 8 sensors-25-05498-f008:**
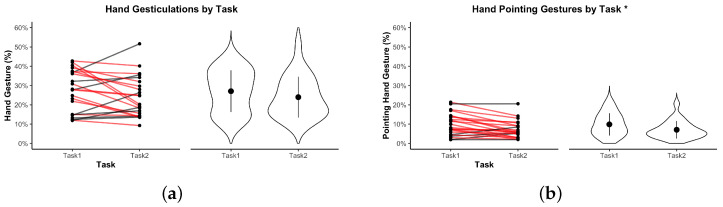
Hand gesture metrics by task. (**a**) Time spent on any expressive gestures. (**b**) Time spent on pointing gestures. Violin plots show distributions. Red lines indicate dyads where the metric decreased from Task 1 to Task 2. * indicates p<0.05.

**Figure 9 sensors-25-05498-f009:**
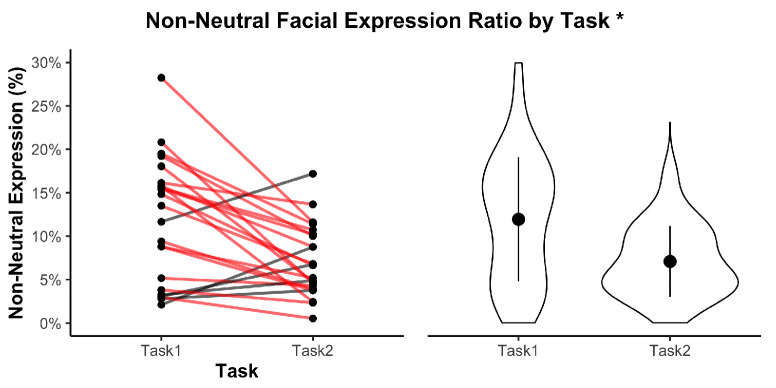
Proportion of non-neutral facial expressions in each task, based on AU intensity thresholds. Violin plots show distributions. Red lines indicate dyads where the metric decreased from Task 1 to Task 2. * indicates p<0.05.

**Figure 10 sensors-25-05498-f010:**
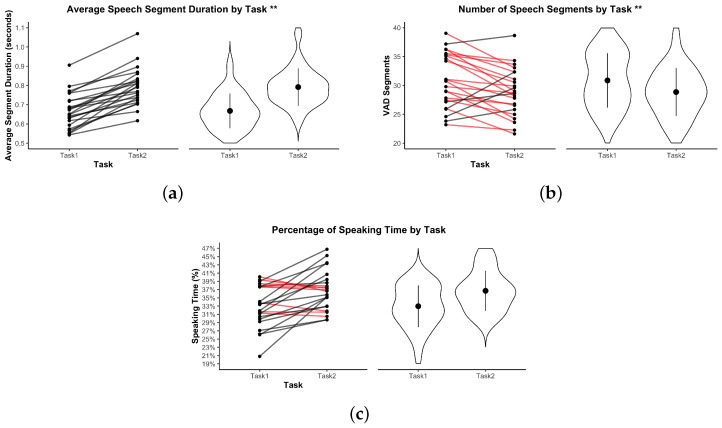
Voice activity metrics by task. (**a**) Average speech segment duration. (**b**) Number of speech segments. (**c**) Percentage of speaking time. Violin plots show variability. Red lines indicate dyads where the metric decreased from Task 1 to Task 2. ** indicates p<0.01.

**Figure 11 sensors-25-05498-f011:**
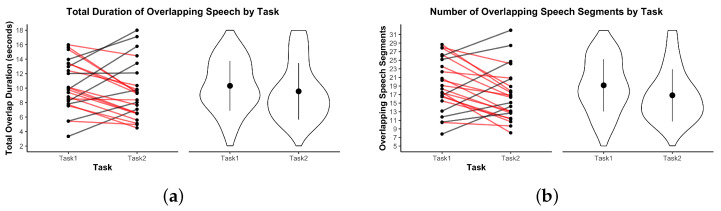
Overlapping speech metrics by task. (**a**) Total overlap duration. (**b**) Number of overlaps. Violin plots show distributions. Red lines indicate dyads where the metric decreased from Task 1 to Task 2.

**Figure 12 sensors-25-05498-f012:**
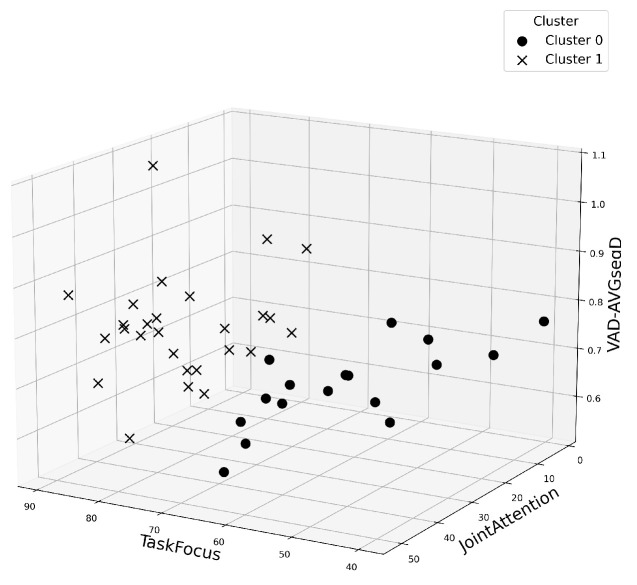
3D clustering of dyads using joint attention, task focus, and average speech segment duration. Shapes indicate cluster membership; K = 2 gave the best separation.

**Table 1 sensors-25-05498-t001:** AU intensity mapping used to approximate discrete FACS annotations.

AU Intensity Range	Qualitative Label
0–0.2	Trace
0.2–0.4	Slight Evidence
0.4–0.6	Marked or Pronounced
0.6–0.8	Severe
0.8–1	Maximum

**Table 2 sensors-25-05498-t002:** Mapping of study hypotheses to corresponding behavioral variables and expected directional effects between tasks.

Hypothesis	Construct	Associated Variables	T1 vs. T2
H1a	Gaze direction (partner-focused)	Eye contact, One-way gaze	T2 > T1
H1b	Gaze direction (task-focused)	Task object focus	T1 > T2
H2	Shared attentional focus	Joint attention	T2 > T1
H3	Physical expressivity	Expressive gestures	T1 > T2
H4	Facial feedback	Non-neutral facial expression ratio	T2 > T1
H5	Speech segments	Speech activity and duration	T2 > T1
H6	Interaction overlap	Overlapping speech segments	T1 > T2
H7	Speech fragmentation	Speech activity and duration	T1 > T2

**Table 3 sensors-25-05498-t003:** Mean proportion and percentage (±SD) of expressive gesture types across tasks.

Gesture	Task 1 Mean	Task 2 Mean	Task 1% ± SD	Task 2% ± SD
Paper	145.74	191.87	9.24 ± 5.25	9.93 ± 7.38
Pointing	209.78	162.65	9.76 ± 5.71	8.44 ± 5.28
Rock	117.17	96.52	7.34 ± 5.61	6.58 ± 4.69
Scissors	1.25	1.43	0.02 ± 0.04	0.02 ± 0.04
Stop	14.47	18.71	0.77 ± 0.99	0.80 ± 0.96
Thumbs-Down	17.09	17.63	1.11 ± 1.20	0.99 ± 1.10
Thumbs-Up	8.31	7.14	0.45 ± 0.79	0.38 ± 0.66

**Table 4 sensors-25-05498-t004:** Cluster assignment versus true task labels.

	Task 1 Count	Task 2 Count
Cluster 0	17	0
Cluster 1	5	22

## Data Availability

The data presented in this study are available on request from the corresponding author due to privacy and ethical restrictions.

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
