# Peer review of "When Action Speaks Louder than Words: Exploring Non-Verbal and Paraverbal Features in Dyadic Collaborative VR"

_sensors, 2025, doi:10.3390/s25175498_

Round 1
Reviewer 1 Report
Comments and Suggestions for Authors
This study investigated how task design in VR shapes non-verbal and paraverbal behaviors during dyadic collaboration. The topic is interesting. Some comments are provided for improving the manuscript quality.
- The English language of the manuscript should be improved.
- The application of VR in difference education contexts should be discussed such as safety training and education.
- The study significance and novelty should be strengthened.
- For the analysis methodology, who to develop the two equations? Any theoretical support?
- The hypotheses in Table 2 should be developed with a literature review.
- More elaborations for each figure should be provided.
Author Response
ROUND 1 REVIEWER RESPONSE
Manuscript Title: Exploring non-verbal and paraverbal features in dyadic collaborative VR
Reviewer 1, Comment 1
“The application of VR in different education contexts should be discussed, such as safety training and education.”
We have added this paragraph to highlight VR’s cross‑industry value while being clear that it is a supplemental tool rather than a one‑to‑one substitute for real‑world training.
New text added in Introduction (Section 1):
" Virtual reality (VR) is increasingly used in education and training for its ability to simulate realistic, risk-free environments. In medical education, VR reduces injuries, speeds up procedures, and improves outcomes by allowing learners to practice safely and repeatedly. Nursing students report that VR is immersive and helpful for building confidence and practicing without risking patient harm. In engineering, students explore complex systems like thermal labs through VR, gaining faster insights than in traditional settings. High-risk industries, such as construction, use VR for safety training. It helps workers memorize hazards and practice procedures in engaging, repeatable ways. These examples show VR’s value in offering embodied, scenario-based learning. Still, VR is best seen as a supplement—not a replacement—for real-world practice, especially for tasks requiring fine motor skills.
Reviewer 1, Comment 2
The study significance and novelty should be strengthened.
We have revised the final paragraph of the Introduction to better emphasize the study’s novel contribution: systematically linking task design features to distinct behavioral patterns across verbal, non-verbal, and paraverbal channels in dyadic VR collaboration.
Revised text added in Section 1.2 (Current Study)
"While prior studies have focused on either communication skills in VR or multimodal tracking separately, this study uniquely explores how specific task constraints—structured versus unstructured, visual versus verbal—systematically shape the emergence of behavioral interaction profiles. Our work highlights how design-driven scenario variation can serve as a scaffold for eliciting and measuring soft skill expression in VR collaboration."
Reviewer 1, Comment 3
“For the analysis methodology, who developed the two equations? Any theoretical support?”
The formulas we used—normalized time ratios and dyad averages—are based on standard analytic practices in studies of multimodal and dyadic interaction. These approaches allow for task-agnostic comparisons and reduce inter-individual variance, which is important when focusing on group-level behavioral dynamics. While not derived from a single theoretical framework, they are commonly used in social signal processing and collaborative behavior research (e.g., Andrist et al., 2013; Vinciarelli et al., 2009).
Clarification added in Section 2.4 (Analysis Methodology)
These formulas follow conventions in multimodal and social interaction research, where behavior normalization and averaging are commonly used to enable comparability across sessions and reduce participant-level variability. Similar strategies have been employed in studies on social signal processing and dyadic collaboration [44,45]
Reviewer 1, Comment 4
“The hypotheses in Table 2 should be developed with a literature review.”
We agree with the reviewer and have revised Section 2.5 (Study Hypotheses) to more clearly ground each hypothesis in relevant theoretical and empirical literature from studies on joint attention, gesture use, facial expression, and turn-taking dynamics in social and collaborative interaction. Each hypothesis remains as originally written but has now been lightly annotated with in-text citations to clarify the theoretical basis for the predicted patterns. Additional references [50-55] have been added to the bibliography, and no changes were made to the hypotheses table itself (Table 2), as it already functioned as a summary.
- For H1 and H2 (gaze and joint attention), we now reference research showing that shared visual spaces reduce the need for gaze-based coordination, while the absence of a shared referent leads to increased reliance on mutual gaze and partner-directed attention.
- For H3 (gestural expressiveness), we added support from gesture studies showing that spontaneous hand movements frequently accompany speech in tasks involving spatial reasoning or physical manipulation, especially in fast-paced contexts.
- For H4 (facial feedback), we now cite research indicating that facial expressions and backchannel cues become especially important in regulating social exchange when explicit referents are unavailable.
- For H5–H7 (paraverbal speech structure), we reference foundational turn-taking literature demonstrating that structured, sequential tasks elicit longer and less fragmented speech turns, while spontaneous, co-active interaction results in more overlap, fragmentation, and rapid turn exchanges.
Reviewer 1, Comment 5
“More elaborations for each figure should be provided.”
We have expanded the figure captions and included clarifying text in the Results section to explain what each figure illustrates, and how to interpret the visual elements.
Section 3.1 Revision
Figure 6a shows the completion time of each team. Seven teams reached the 16-minute time cap in Task 1, compared to only five in Task 2. No significant difference in overall task duration was observed between Task 1 (M = 12.21min, SD = 4.20min; ≈ 12 minutes 13 seconds) and Task 2 (M = 13.65min, SD = 2.68min; ≈ 13 minutes 39 seconds), t(22) = 1.40, p = .1764, Bonferroni-corrected p = 1.000, d = 0.29.
Section 3.2 Revision
Figure 6a shows the average proportion of partner-directed gaze across both tasks. In Task 1, teams displayed an average of 2.48% partner-directed gaze (SD = 2.16), corresponding to 18.17 ± 15.82 seconds of eye contact during a mean task duration of 12 minutes and 13 seconds. In Task 2, the average dropped to 1.46% (SD = 1.37), equivalent to 11.96 ± 11.16 seconds, with a slightly longer mean duration of 13 minutes and 39 seconds. While a paired-samples t-test indicated a nominally significant difference between conditions (t(22) = -2.08, p = .0497), this effect did not survive Bonferroni correction (p = .9438). Taken together, eye contact accounted for only a small fraction of the interaction in both tasks—approximately 12–18 seconds across 12–14 minutes—and did not differ robustly after correction. Variability in eye contact was also greater in Task 1, indicating more between-team heterogeneity in partner-directed gaze under dynamic, parallel collaboration.
Figure 6b displays the proportion of gaze directed at task-relevant elements. In Task 1, participants spent on average M = 64.56% (SD = 9.76) of their viewing time on task objects, corresponding to approximately 7 minutes 53 seconds (± 1min 11s) of the average duration. In Task 2, this increased to M = 78.90% (SD = 5.51), or roughly 10 minutes 46 seconds (± 45s) of the average task duration. This difference was highly significant, t(22) = 8.83, p < .0001, and remained so after Bonferroni correction (p < .0001), with a large effect size (d = 1.84). The higher proportion in Task 2 suggests that its structured, turn-based coordination anchored participants’ visual attention more consistently to relevant objects, whereas Task 1’s dynamic pacing and shared visual field allowed more gaze to drift toward non-task elements.
Joint attention duration is illustrated in Figure 6c. In Task 1, participants spent on average M = 20.13% (SD = 10.95) of the trial in joint attention, corresponding to approximately 2 minutes 27 seconds (± 1min 20s) of the average task duration. In Task 2, this increased to M = 38.07% (SD = 8.59), or roughly 5 minutes 12 seconds (± 1min 10s) of the average task duration. This difference was statistically significant, t(22) = 9.00, p < .0001, and remained so after Bonferroni correction (p < .0001), with a strong effect size (d = 1.88). The marked increase in Task 2 aligns with its design constraints, which required frequent coordinated reference to the same conceptual target despite participants not sharing a physical view of each other’s objects.
Figure 6d presents joint attention frequency. In Task 1, teams aligned their visual focus an average of M = 69.57 times (SD = 38.65) during the 12 minutes 13 seconds average task duration. In Task 2, this increased to M = 105.43 episodes (SD = 28.27) over the 13 minutes 39 seconds average task duration. The difference was statistically significant, t(22) = 3.62, p = .0015, and remained significant after Bonferroni correction (p = .0287), corresponding to a medium-to-large effect size (d = 0.76). The higher frequency in Task 2 reflects the need for continual verification of shared focus when no visible common workspace is available, requiring more frequent gaze-based coordination.
Changes to Captions
- Figure 6: Task performance across Task 1 and Task 2. (a) Task duration per team, showing those hitting the 16-min cap. (b) Completion rates per team. Violin plots show distribution.
- Figure 7: Eye-gaze metrics for Task 1 vs. Task 2. (a) Eye contact ratio. (b) Task object focus. (c) Joint attention duration. (d) Joint attention frequency. Violin plots show distributions.
- Figure 8: Hand gesture metrics by task. (a) Time spent on any expressive gestures. (b) Time spent on pointing gestures. Violin plots show distributions.
- Figure 9: Proportion of non-neutral facial expressions in each task, based on AU intensity thresholds. Violin plots show distributions.
- Figure 10: Voice activity metrics by task. (a) Average speech segment duration. (b) Number of speech segments. (c) Percentage of speaking time. Violin plots show variability.
- Figure 11: Overlapping speech metrics by task. (a) Total overlap duration. (b) Number of overlaps. Violin plots show distributions.
- Figure 12: 3D clustering of dyads using joint attention, task focus, and average speech segment duration. Shapes indicate cluster membership; K = 2 gave the best separation.
Additional Note on Revisions
In response to reviewer feedback, targeted adjustments have been made throughout the manuscript to improve English language use, grammar, and overall readability. These changes include streamlining sentence structure, ensuring terminology consistency, and enhancing the clarity of methodological descriptions.
Reviewer 2 Report
Comments and Suggestions for Authors
Well-structured article and sound methodology.
I would recommend reducing the number of Hypothesis and also include questions related to limitations of the study.
Author Response
ROUND 1 REVIEWER RESPONSE
Manuscript Title: Exploring non-verbal and paraverbal features in dyadic collaborative VR
Reviewer 2, Comment 1
“Reduce number of hypotheses.”
We thank the reviewer for this helpful suggestion. We have chosen to retain all seven hypotheses, as each reflects a distinct aspect of the multimodal communication dynamics we aim to assess (gaze, gestures, facial cues, speech timing, etc.). Reviewer 1 did not raise concerns regarding the number of hypotheses, and we believe that addressing these distinct channels separately is important for interpretability. Each hypothesis is now explicitly grounded in relevant theoretical literature (see Section 2.5), and their measurement mapping is clearly summarized in Table 2 to maintain clarity.
Reviewer 2, Comment 2
“Include questions or discussion about study limitations.”
We have added a Limitations and Future Directions subsection (Section 4.1) addressing avatar realism, participant familiarity, hand-tracking constraints, absence of subjective measures, and sample size. This section outlines the study’s boundaries and proposes clear directions for future research.
4.1 Limitations and future directions
Several limitations of our study warrant consideration. First, although our analyses focused on non-verbal and paraverbal behavior, these interactions were mediated through simplified avatars. The fidelity of eye-gaze, facial expression, and hand tracking in current VR systems remains limited, which can constrain how well partners interpret each other’s intentions. Prior work in social VR has shown that greater avatar realism and more expressive non-verbal channels can enhance co-presence and the interpretability of social cues, thereby influencing interaction outcomes [60-62]. Our findings should therefore be interpreted as indicative of relative differences between task conditions rather than as direct proxies for real-world behavior.
Second, our sample comprised predominantly familiar dyads. Familiarity may facilitate coordination and reduce communicative effort, limiting the generalization of our findings to strangers or mixed-expertise teams. Although this recruitment bias is noted in the Participant section, future work should systematically compare familiar and unfamiliar dyads to assess how relationship dynamics influence non-verbal and paraverbal behavior.
As previously stated, we observed technical constraints related to hand tracking. Because participants relied on optical hand tracking to grasp and place virtual objects, imprecise tracking occasionally led to frustration and disrupted task flow. In the post-session debriefing, 31 participants reported more difficulty manipulating objects in Task 2 (Structured, Unknown Goal), where precise placement was required. We recommend that future studies use controller-based manipulation or improved tracking systems when object handling is central to the task. We did not include formal measures of user comfort or quality of experience. While participants informally reported feeling more strain during precise object manipulation, future experiments should incorporate validated subjective measures (e.g., presence, workload, frustration) to better understand how VR task design influences both experience and the interpretation of partner behaviors.
Finally, our study was limited to two tasks with specific structural properties (dynamic versus structured, known versus unknown goal) and a sample size of 23 dyads, which was adequate for the repeated-measures comparisons at the dyad level reported here. However, human interaction is inherently complex and multimodal, and future work should incorporate a broader range of tasks varying in complexity, role symmetry, and feedback modality, alongside substantially larger datasets. Such expansions would allow for robust testing of the generalizability of our findings and enable multilevel analyses that capture both individual- and dyad-level variation, providing a more nuanced understanding of how behavioral patterns emerge and differ across contexts.
Additional Note on Revisions
In response to reviewer feedback, targeted adjustments have been made throughout the manuscript to improve English language use, grammar, and overall readability. These changes include streamlining sentence structure, ensuring terminology consistency, and enhancing the clarity of methodological descriptions.
Reviewer 3 Report
Comments and Suggestions for Authors
The paper presents an interesting issue about evaluating nonverbal and paraverbal emotions.
However, the article suffers from three major shortcomings that compromise its technical and ecological validity.
First, it does not model the user-avatar relationship, failing to describe how real-life actions (gestures, gazes) are translated into mediated collaborative interactions, particularly via visual/haptic indicators allowing partners to interpret the intentions behind avatar movements.
Second, the recruitment of 87.5% familiar dyads (pairs of friends) introduces an uncontrolled bias, distorting the observed collaborative dynamics without comparative analysis with unfamiliar dyads.
Third, the absence of subjective data (perceived co-presence, technical frustration) and the neutralization of individual expressions by standardized avatars limit its transferability to real-world contexts. These omissions prevent us from assessing whether the measured behaviors reflect authentic social skills or device artifacts, particularly in the face of documented but unquantified technical issues (imprecise object grasping, network latency).
A revision therefore requires: (1) explanatory diagrams of the sensor → network → rendering technical flow, (2) the exploitation of unfamiliar dyads to isolate the effect of familiarity, and (3) the integration of user perception measures.
In summary, the authors should revise Section 2.3. Indeed, they will need to explain in detail the relationship between the behavior of real users and their corresponding avatars. Subsequently, it is necessary to deeply describe how paraverbal and nonverbal behaviors were observed and analyzed in real users, not only on avatars.
Comments on the Quality of English LanguageThe quality of the English language could be improved.
Author Response
ROUND 1 REVIEWER RESPONSE
Manuscript Title: Exploring non-verbal and paraverbal features in dyadic collaborative VR
Reviewer 3, Comment 1
“The paper does not model the user-avatar relationship, failing to describe how real-life actions are translated into mediated collaborative interactions, particularly via visual/haptic indicators allowing partners to interpret the intentions behind avatar movements.”
We thank the reviewer for this suggestion. While Section 2.2 (Apparatus) already describes the devices, SDK, and tracking modalities used to translate participants’ real-world movements (eye, face, head, and hands) into avatar behaviors, we have now expanded this section to more explicitly outline the technical flow from sensor capture to avatar rendering in the partner’s view. This includes details of the sampling rates, processing via the Meta SDK, real-time network synchronization using Photon, and Unity-based avatar animation.
To further clarify this process, we have added a new figure (Figure 1) illustrating the sensor → processing → network → rendering pipeline for both participants in the dyadic setup, showing how each user’s movements are captured, transmitted, and displayed on their partner’s avatar in real time. This figure complements the textual description and makes the data flow between real-world actions and mediated interactions explicit.
Reviewer 3, Comment 2
“The recruitment of 87.5% familiar dyads introduces bias, distorting the observed collaborative dynamics without comparative analysis with unfamiliar dyads.”
We acknowledge this recruitment bias and now explicitly address it in the Limitations and Future Directions section (Section 4.1). We note that the predominance of familiar dyads may have influenced coordination ease and non-verbal communication, potentially reducing the generalizability to unfamiliar or mixed-expertise pairs. Future work is recommended to systematically compare familiar and unfamiliar dyads to better isolate the role of relationship dynamics.
Reviewer 3, Comment 3
“The absence of subjective data (perceived co-presence, technical frustration) and the neutralization of individual expressions by standardized avatars limit transferability to real-world contexts.”
We agree and have added this as a study limitation in Section 4.1. We highlight that while our focus was on relative differences between task conditions, we did not formally collect subjective experience measures. Informal debriefings indicated that 31 participants found object manipulation more difficult in Task 2. We recommend including measures such as presence, workload, and frustration in future studies to better understand user experience. Looking forward, an important research direction is to predict subjective experience—such as perceived co-presence, satisfaction, or indicators of successful collaboration—directly from multimodal sensor data. This would reduce reliance on self-report, allow real-time feedback, and potentially improve the ecological validity of VR collaboration assessments.
Reviewer 3, Comment 4
“Explain in detail the relationship between the behavior of real users and their corresponding avatars.”
We clarify that the scope of this study is not to validate VR behaviors against real-world counterparts, but to investigate VR as an interaction medium in its own right. While both scenarios were initially prototyped in a physical setting to assess task feasibility, the presented results are based solely on VR interaction data, interpreted within the constraints and affordances of the VR environment.
Additional Note on Revisions
In response to reviewer feedback, targeted adjustments have been made throughout the manuscript to improve English language use, grammar, and overall readability. These changes include streamlining sentence structure, ensuring terminology consistency, and enhancing the clarity of methodological descriptions.
Reviewer 4 Report
Comments and Suggestions for Authors
This study presents a promising application of VR-based assessment; however, several limitations should be more thoroughly addressed. The approach remains heavily dependent on human experts to interpret behavioral data, which may introduce potential bias and subjectivity. Furthermore, the high cost and technical complexity associated with advanced VR hardware and tracking systems continue to pose substantial barriers to broader adoption, particularly in real-world or resource-constrained settings.
While VR technology can effectively capture a range of behavioral responses, the simulation and interpretation of subtle social cues and culturally specific nuances remain challenging. The authors may also wish to consider how user comfort and adaptability influence assessment outcomes, as individual differences in acclimatization to virtual environments could affect data reliability. Lastly, despite a high degree of realism, VR-based scenarios may still fall short in replicating the full complexity and ecological validity of real-world social interactions, thereby limiting the generalizability of the observed behaviors.
Author Response
ROUND 1 REVIEWER RESPONSE
Manuscript Title: Exploring non-verbal and paraverbal features in dyadic collaborative VR
Reviewer 4, Comment:
“The study presents a promising application of VR-based assessment; however, several limitations should be more thoroughly addressed, including potential bias from human expert interpretation, technical complexity, user comfort, and ecological validity.”
We thank the reviewer for these valuable observations. Section 4.1 (Limitations and future directions) has been substantially expanded to address them. The revised text now elaborates on how the interpretation of behaviors is constrained by the use of simplified avatars and current tracking fidelity, drawing on literature concerning avatar realism and the perception of social cues. We also clarify that our sample consisted predominantly of familiar dyads, which may have influenced coordination patterns, and note the implications for generalizability. Technical constraints, particularly the limitations of optical hand-tracking and participant-reported frustration during precise object manipulation, are discussed alongside recommendations for alternative input methods. We further acknowledge the absence of formal subjective measures—such as presence, workload, and comfort—and outline the importance of incorporating these in future studies. The section now also reflects on sample size, noting that larger datasets will be necessary to conduct multilevel analyses capable of capturing the full complexity of human interaction. Finally, we briefly address the scalability of our approach, recognizing that the cost and technical requirements of high-end VR systems may limit adoption in certain applied contexts. These additions reinforce that our findings should be interpreted as patterns emerging within VR-specific scenarios shaped by task design, rather than as direct analogues of real-world interactions.
Additional Note on Revisions
In response to reviewer feedback, targeted adjustments have been made throughout the manuscript to improve English language use, grammar, and overall readability. These changes include streamlining sentence structure, ensuring terminology consistency, and enhancing the clarity of methodological descriptions.
Round 2
Reviewer 1 Report
Comments and Suggestions for Authors
The authors have addressed my comments properly.